# An open platform for visual stimulation of insects

**Stefan Prech** [1]*, **Lukas N. Groschner** [1,2], **Alexander Borst**[1]

**1** Max Planck Institute for Biological Intelligence, Martinsried, Germany, **2** Gottfried Schatz Research Center, Molecular Biology and Biochemistry, Medical University of Graz, Graz, Austria

* stefan.prech@bi.mpg.de

**Data Availability Statement:** Data and code are available at https://github.com/borstlab/super_bowl_screen.

**Funding:** The Max Planck Society (https://www.mpg.de). Support for SP, LNG and AB. Horizon

## Abstract

To study how the nervous system processes visual information, experimenters must record neural activity while delivering visual stimuli in a controlled fashion. In animals with a nearly panoramic field of view, such as flies, precise stimulation of the entire visual field is challenging. We describe a projector-based device for stimulation of the insect visual system under a microscope. The device is based on a bowl-shaped screen that provides a wide and nearly distortion-free field of view. It is compact, cheap, easy to assemble, and easy to operate using the included open-source software for stimulus generation. We validate the virtual reality system technically and demonstrate its capabilities in a series of experiments at two levels: the cellular, by measuring the membrane potential responses of visual interneurons; and the organismal, by recording optomotor and fixation behavior of *Drosophila melanogaster* in tethered flight. Our experiments reveal the importance of stimulating the visual system of an insect with a wide field of view, and we provide a simple solution to do so.

## Introduction

Neural responses are inherently variable. The visual system is no exception in this respect, as repeated presentations of the same stimulus tend to elicit different neural responses [1–3]. To decipher the influences of visual stimulation on neuronal responses, experimenters must be able to deliver light to the eye in a controllable and reliable manner. An ideal visual stimulation device would provide precise temporal and spatial control over every retinal photoreceptor cell. In recent years, research on the visual system of insects, in particular that of *Drosophila melanogaster*, has produced a rich collection of stimulation devices that take many shapes and forms. Fueled by technological advances, most of them either repurpose devices originally designed for the human visual system [4–9], or are custom-made from highly specialized components [10–14] rendering their implementation almost prohibitively expensive. To tailor a stimulation device to the demands of an insect's visual system, while keeping it versatile, compact, and affordable, poses a formidable technical challenge.

In contrast to the human eye, the compound eye of *Drosophila* is tiled by hundreds of ommatidia arranged in a hexagonal grid. Each ommatidium of the convex retina collects light from a visual angle of approximately 5° using a miniature lens with an aperture of 16 μm and a

2020 programme under the Marie Skłodowska-Curie Action MOVIS (https://marie-sklodowska-curie-actions.ec.europa.eu, grant agreement no. 896143). Support for LNG. The funders had no role in study design, data collection and analysis, decision to publish, or preparation of the manuscript.

**Competing interests:** The authors have declared that no competing interests exist.

focal length of 20 μm [15–18]. These dimensions, for the most part, obviate the need to accommodate, as any object beyond a few millimeters appears at infinity focus. With an inter-ommatidial angle of ~5˚, the spatial resolution of *Drosophila* is low, but the convex geometry of its eyes provides for a nearly panoramic field of view [19, 20].

Dynamic visual environments for head-restrained insects that allow experimenters to record behavioral and neural responses to visual stimuli are commonplace in insect neuroethology [21]. Relics of long-gone manufacturing constraints, most displays used to create these virtual environments have either a flat or a cylindrical geometry, neither of which matches the shape of the fly eye. Hence, they all bear at least three limitations: First, they cover only a small fraction of the fly's field of view; second, the brightness of the display varies across visual space as a function of the viewing angle; and third, distortions in the periphery of the visual field complicate the correspondence between actuator (pixels) and sensor (ommatidia) (Fig 1A–1C). While the latter two can be compensated for by computationally costly pre-adjustment and pre-distortion of the displayed images, the first caveat is unavoidable.

Here, we describe an inexpensive projector-based visual stimulation device that caters to many of the demands of the insect visual system. It provides a wide and nearly distortion-free field of view with uniform brightness, but is compact enough to fit under a microscope. The crux of our system lies in the unconventional screen geometry, shaped like a quarter of a football, which simplifies both software and hardware development. Our open-source Python code allows any user with access to a computer and a 3D printer to create a bowl-shaped screen that suites the specifications (i.e. the throw ratio) of their projector. The temporal, spatial and spectral performance of the system can be adjusted, according to experimental requirements and budget constraints by resorting to different commercially available projectors. We provide Python code to generate the screen design and to export it for 3D printing, but also to display a wide repertoire of visual stimuli commonly used by researchers, teachers, and students who study the insect visual system.

## Results

### Comparing screen geometries

To simulate and quantify the perspective distortions that occur when using different screen geometries, we used a polyhedron as a general, simplified model of the invertebrate compound eye. The screen capture from each projected polyhedron face can be understood as the number of projector pixels required to stimulate a certain fraction of the visual field, for example, one ommatidium or one spatial receptive field of a neuron. To allow direct comparison of different screen geometries, a Goldberg polyhedron containing 272 almost identical faces served as a model observer in front of a flat, a cylindrical, or a bowl-shaped virtual surface (Fig 1A–1C).

Observing the grid of the polyhedron projected onto a flat surface (Fig 1A), only the hexagon at the center of the visual field appeared undistorted. With increasing distance from the center, the hexagons grew rapidly in surface area and were increasingly distorted. At an angular distance of 50˚ from the center, hexagonal projections were on average 3.3 times longer and 1.9 times wider, after correcting for the hexagon's original aspect ratio; their area increased by a factor of 5.5. On a cylindrical surface (Fig 1B), hexagons remained undistorted along the equator but were subject to heavy distortions along the elevation. At an angular distance of 50˚ from the center, the average hexagon grew by a factor of 2.9 in elevation and 1.6 in azimuth. The area of a hexagonal projection increased by an average factor of 4.6. The bowl-shaped surface featured no distortion along the elevation and only slight azimuthal distortions in the periphery of the screen (Fig 1C). At an elevation angle of 50˚ from the pole, the azimuthal distortion factor amounted to 1.2; at 100˚ from the pole, it was 2.4. The area of the hexagonal

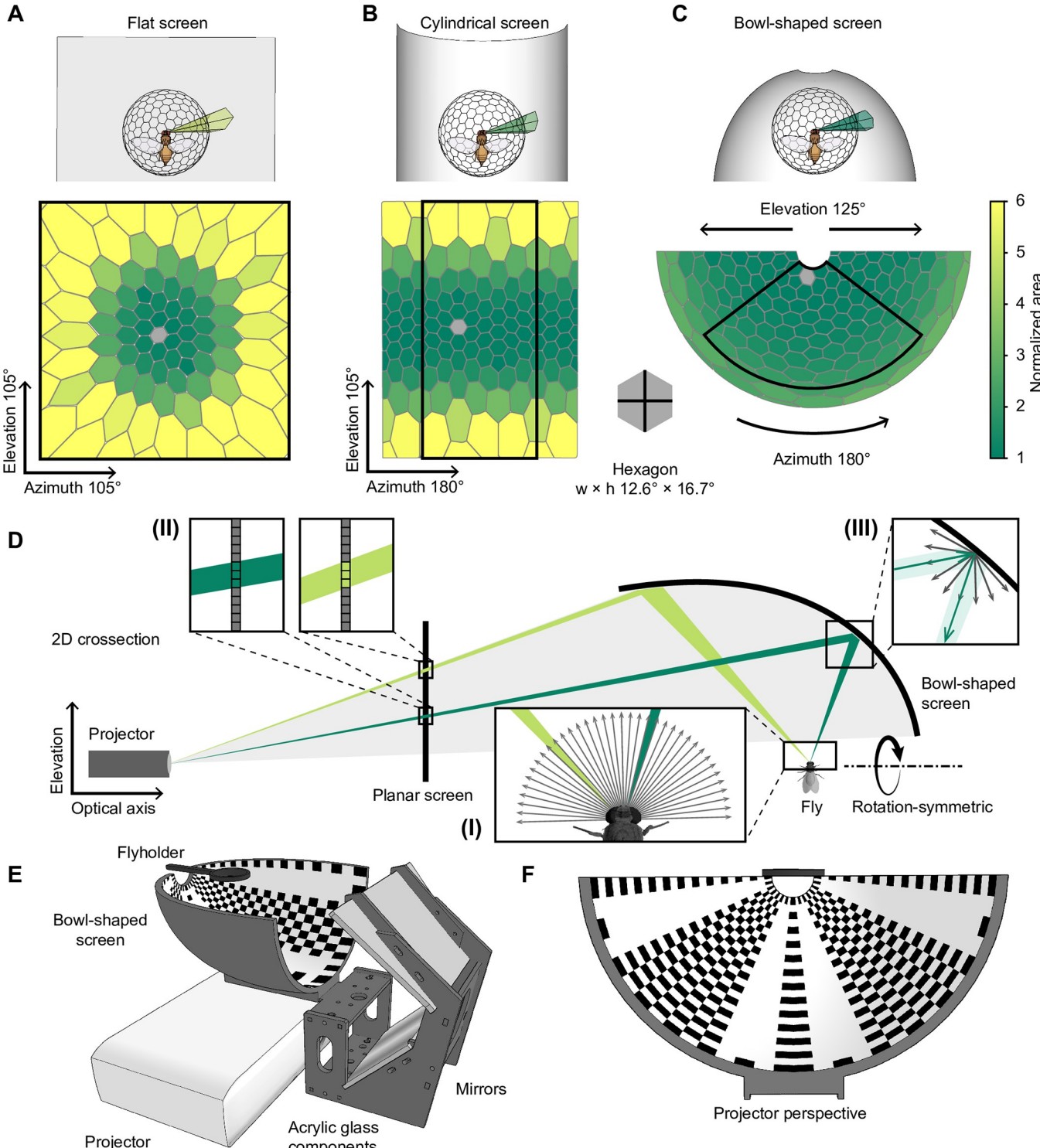

**Fig 1. Geometry of the bowl-shaped screen. A-C** Renderings of a virtual projection of a hexagonal grid onto a planar (A), a cylindrical (B), and a bowl-shaped screen (C). The colors correspond to area sizes and are normalized to a standard undistorted hexagon at the center (gray). One hexagon corresponds to the visual space covered by approximately nine ommatidia of *Drosophila melanogaster*. **D** Cross-section of the model used to calculate the screen geometry (Eq 3). (I) Spherical photoreceptor arrangement of the fly, assuming a constant inter-ommatidial angle of 5°. (II) Projector designed to produce a uniform illumination and resolution on a planar screen. (III) The shape of the screen and its Lambertian reflection properties guarantee homogeneous spatial luminance distribution across the entire surface. **E** View of the bowl-shaped screen, the mirrors and the projector. **F** Projector perspective of the screen with the necessary equidistant azimuthal projection of the pattern in E.

projection increased by the same factor of 1.2 at 50˚ and 2.4 at 100˚ from the center. Aside from robustness against distortions, a bowl-shaped screen features another advantage: It covers a substantially larger fraction of the visual field compared to a flat or a cylindrical screen of the same surface area. A cylindrical shape requires 44% of the area to cover the same visual field as a flat screen. The bowl-shaped screen can stimulate the same field of view with only 24% of the screen surface.

## Screen design and evaluation

Combining an image projector and a bowl-shaped screen, forms the basis of the Super Bowl display system. Like in other virtual reality systems, the image must be pre-distorted in order to be accurately perceived from the position of the observer. Both the observer and the projector are stationary, which allows us to account for any perspectival distortions in a single step. The correction is not based on a computationally intensive software solution, as is usually the case, but on a specific hardware configuration that uses a precalculated screen geometry at no computational costs. To determine the ideal screen shape, we created models of both the observer and the projector. Here, a sphere with a polar grid served as an approximation of the compound eye, where the optical axis of each ommatidium was represented by a radial vector connecting the center of the sphere with the center of the respective face. Taking advantage of the rotational symmetry of the sphere, we reduced the model of the observer to a two-dimensional cross section (Fig 1D I) (Eq 1). The projector model was simplified to the same extent by considering only two dimensions: one along the optical axis of the projector and a second one along the elevation (Fig 1D). Commercially available projectors are designed to project images with uniform pixel size and uniform luminance onto a planar screen (Fig 1D II) (Eq 2). To guarantee uniform brightness across the visual field of the insect, it is necessary to maintain a constant ratio of pixels (photons) per ommatidium of the convex retina (Fig 1D). A precise correspondence between computer-generated pixels and the retinotopy of the insect visual system can only be established if this ratio is constant across the field of view. We implemented this by intersecting the optical axis vector of each ommatidium along the elevation plane with the modeled photon vector originating at the projector. The trajectory connecting the resulting intersection points defined the screen curvature along the elevation (Fig 1D) (Eq 3). The curvature of the screen along the azimuth was obtained by rotating the trajectory about the polar axis (Eq 4). The resulting concave surface was predicted to be of uniform brightness, from the insect's point of view, provided that the screen surface features diffuse, Lambertian reflectance independent of the incidence angle (Fig 1D III). To achieve this property, and to avoid artefacts caused by a cascade of specular reflection (Fig 2A), we coated the inner surface of the 3D-printed screen with different combinations of white paint and varnish. In order to test the optical properties of the surface, we illuminated square spots (5˚ × 5˚) along the elevation and placed a wide-angle camera (Fusion, GoPro) with a panoramic 360˚ field of view at the position of the observer to record the reflected light (Fig 2B and 2C). One particular combination of paint and varnish (see Methods) showed virtually no specular reflection along the elevation and was used in all subsequent experiments (Fig 2B and 2C).

Luminance measurements from the position of the observer using a rotatable power meter, as well as a 360˚ camera, corroborated our prediction and revealed a nearly isotropic luminance across the entire elevation of the screen (Fig 2D and 2E). The calculated brightness uniformity (i.e., the lowest luminance reading at any point as a fraction of the highest) amounted to 73%, which is comparable to the uniformity other projectors achieve on a planar screen. For instance, the Texas Instruments DLP® LightCrafter™ projectors 4710 EVM or 3000 EVM,

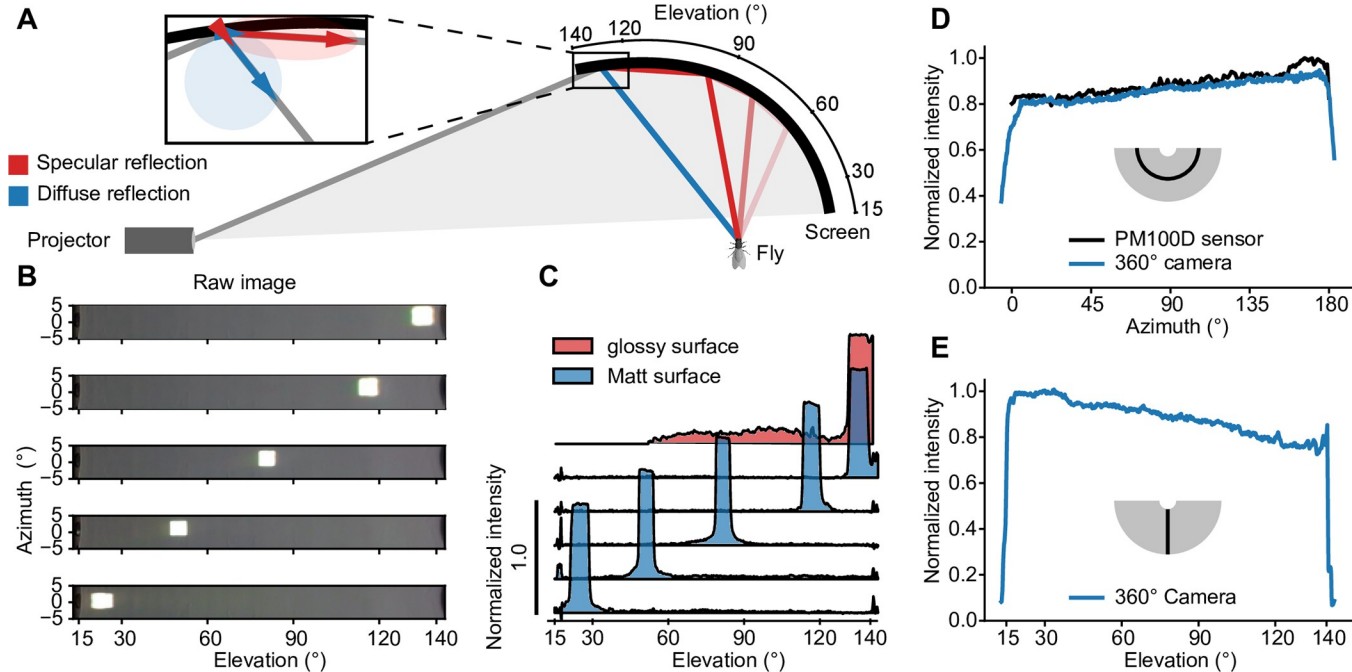

**Fig 2. Technical validation of the operating principle. A** Overview of Lambertian and specular reflectance properties of the screen surface. The blue vectors describe the diffuse, the red vectors the specular reflection component of an entering light ray. Flat angles of incidence cause a reflective cascade on the screen surface. **B** Raw images of bright rectangular squares displayed at different locations along the elevation. **C** Luminance profile along the elevation of a glossy (red) and a matte (blue) surface. The matte surface was evaluated with bright squares projected onto different locations along the elevation. In contrast to the matte surface, the uncoated, glossy surface showed a pronounced reflection cascade. **D, E** Luminance profiles along the azimuth (D) and the elevation (E), measured with a 360˚ camera (blue). Measurements along the azimuth were validated using an optical power meter (black).

which are used in several stimulation systems, reach about 70% brightness uniformity [4, 6, 22].

The bowl-shaped screen was held in place by a scaffold made of acrylic glass components (Fig 1E). Two mirrors served to reduce the overall size of the set-up, but could be omitted depending on spatial constraints. In the present configuration, the mirrors allowed for the projector to be placed underneath the screen. In contrast to other virtual reality systems the projector is easily replaceable and maintainable, making the system to the largest possible extent independent of the rapid product turn over on the consumer market. We used a standard projector (LG PH510PG) with a native resolution of 1280 × 720 pixels, a throw ratio of 1.4, a projection offset of 100% upwards, and an intrinsic brightness uniformity of ~90%. The projector had an image refresh rate of 60 Hz and a flicker frequency of 240 Hz, which exceeds the critical flicker fusion frequency of most insects including that of *Drosophila* [23]. Higher refresh rates could be achieved by using projectors like the ViewSonic® M2E (120 Hz) or the Texas Instruments DLP® LightCrafter™ 4500 EVM (400 Hz).

## Stimulus design and validation

In order to create an immersive, panoramic virtual reality for the animal, we take the insect observer to be at the center of a fictive unit sphere from where it observes an equidistant azimuthal projection of the sphere. Owing to the special geometry of the screen, this projection is —conveniently—identical to the image that must be displayed by the projector (Fig 1F). In other words, the screen geometry allows the direct projection of a spherical texture. The equidistant azimuthal projection is formally equivalent to a polar-transformed version of an

equirectangular projection, the latter of which might be more familiar to the reader, for example, as a Cartesian map of the Earth. These interpretable equirectangular projections served as spherical textures and were chosen as the input format. In order to avoid subsequent interpolation, the projectors polar coordinate system was transformed into the Cartesian coordinate system of the input texture using two precalculated transformation matrices: one for the x- and another for the y-coordinate of each pixel. The size of each matrix, and therefore the runtime, depended on the spatial resolution of the projector. Since the output image size, the transformation matrices, and the input image size were known (and constant) during runtime, we used just-in-time compilation within the JAX (Google) framework to calculate the transformations on the computer's graphics processing unit (GPU). On a standard laptop with a GTX1060 (Nvidia) graphics card, the transformation of a color image of $1280 \times 720$ pixels took on average 2–3 ms. Before the transformation, to speed up processing times, equirectangular representations were cropped to the field of view covered by the screen (here, $180° \times 125°$ in azimuth and elevation, respectively) and stimuli with a rotational component were pre-rotated (Fig 3B). To avoid direct exposure of the fly to distracting projector beams, and to reduce stray light in the surroundings, a mask was applied to illuminate only the screen. This mask could be adapted as required to incorporate additional elements such as timestamp signals or cameras for monitoring behavior.

To put our stimulation device to a test, we used the panoramic 360° camera. From the position of the observer, the camera provided a high-resolution equirectangular projection of the entire screen, which—given correct transformation and alignment—should be identical to the input image. Comparison between the two allowed for a qualitative assessment of possible image distortions. As an image, we used the panorama from inside a cube with differently colored sides (Fig 3A) and inspected the transformed image at each step in the process chain: the cropped equirectangular projection, which served as the input image (Fig 3B), the projected image (Fig 3C), and the image captured by the panoramic camera (Fig 3D). The cropped input image (Fig 3B) and the observed image (Fig 3D) were virtually identical in terms of their proportions, grid geometry, contrast, sharpness, and field of view. This attests to the accuracy of the transformation and the correct, undistorted display of images on the 3D-printed screen.

Time-varying visual stimuli were generated using custom-written software in Python. To assess the temporal performance, rotational stimuli, like sine wave gratings moving at various velocities (Fig 3E), and online-generated stimuli, such as a discrete white noise stimulus (Fig 3F), were evaluated using a fast photodiode as a fly's proxy. In case of moving sine wave gratings, there were no conspicuous flickers in the down-sampled photodiode signal (<120 Hz), which might be perceived by insects. The amplitude, the frequency and the phases of the signals were identical for all velocities (Fig 3E). Before testing our bowl-shaped screen on real neurons with uncertain temporal filter properties, we measured the temporal precision of our stimulation device by measuring the spatiotemporal "receptive field" of a photodiode for reference. We projected onto the screen a 60 Hz spatiotemporal binary white noise stimulus over a period of 8 min. Cross-correlating the luminance at each screen position with the photodiode signal revealed a positive correlation confined to the first 3 frames (50 ms). This sharply-tuned temporal receptive field suggests that, over the stimulation period of 8 min, jitter and drift are in the range of ±1 frame (16.67 ms) (Fig 3F).

The bowl-shaped screen makes it possible to stimulate the visual system in an identical manner at various points in space. This is essential, for instance, when spatiotemporal receptive fields of neurons at different spatial locations in the visual system are to be measured precisely. To test the spatial precision of our system, we measured the spatial "receptive fields" of the same photodiode placed at two different locations in the arena (Fig 4A) and corrected the resulting equirectangular image of the receptive field by rotating the position to the

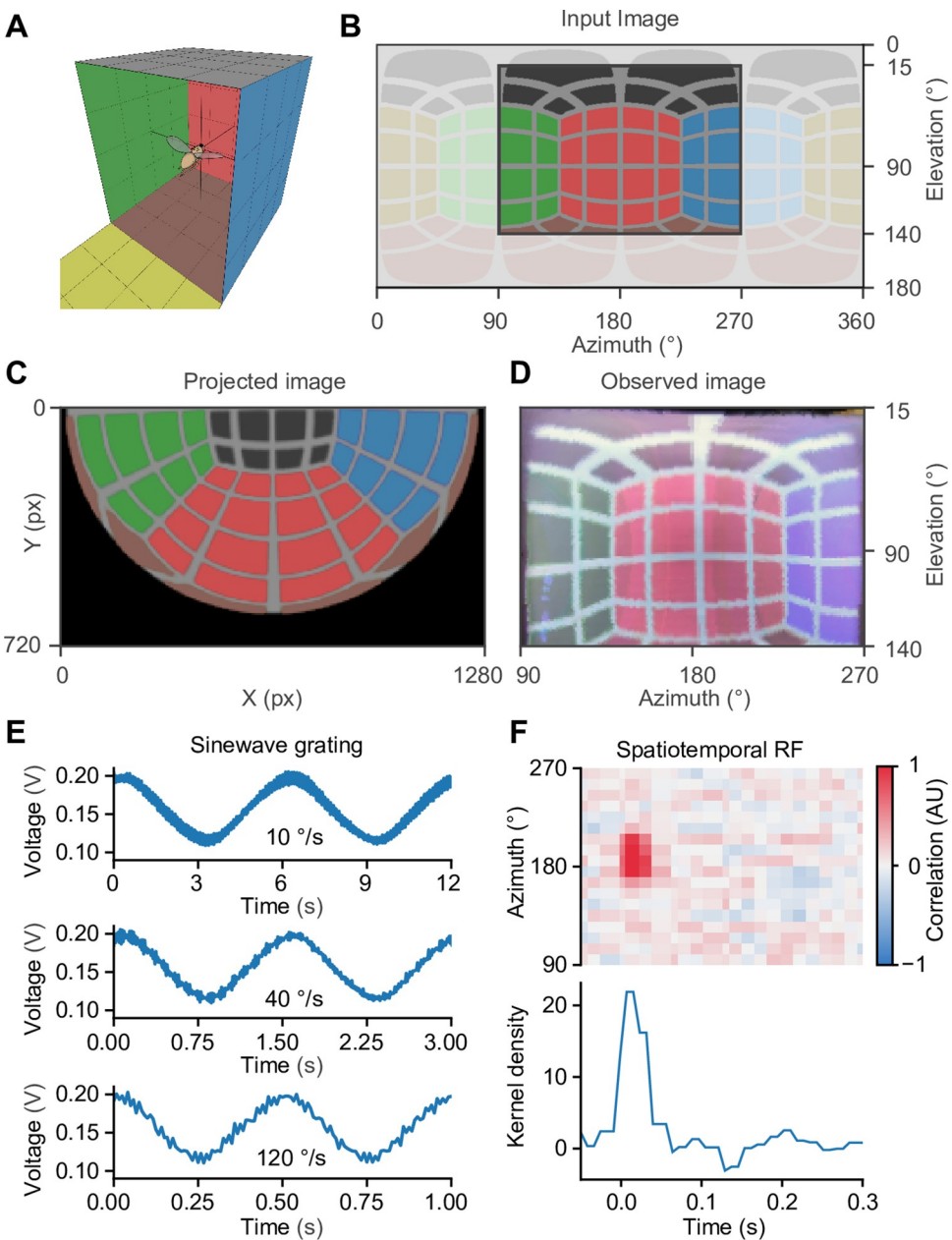

**Fig 3. Technical validation of spatial and temporal stimulus control. A** 3D view of the fly inside a virtual cube with colored walls. **B** Input image, an equidistant cylindrical projection of the virtual cube. The entire image is shown in light color, the image displayed on the screen is shown in dark color. **C** Image from the viewpoint of the projector to be displayed on the bowl-shaped screen. **D** Captured image of the bowl-shaped screen from the 360˚ camera in equidistant cylindrical projection, the direct comparison to the dark region in B. **E** Sinewave gratings moving across the screen at three different velocities captured by a photodiode and resampled to a framerate of 120 Hz. **F** Spatiotemporal "receptive field" of the photodiode, measured over 8 minutes.

undistorted center of the spherical texture (Fig 4B). The result demonstrates that the perspective correction yielded identical receptive field sizes, even in cases where the receptive field was located more than 50˚ off the center. The occurring deviations of about 8% were below the resolution of the binary white noise grid (Fig 4B).

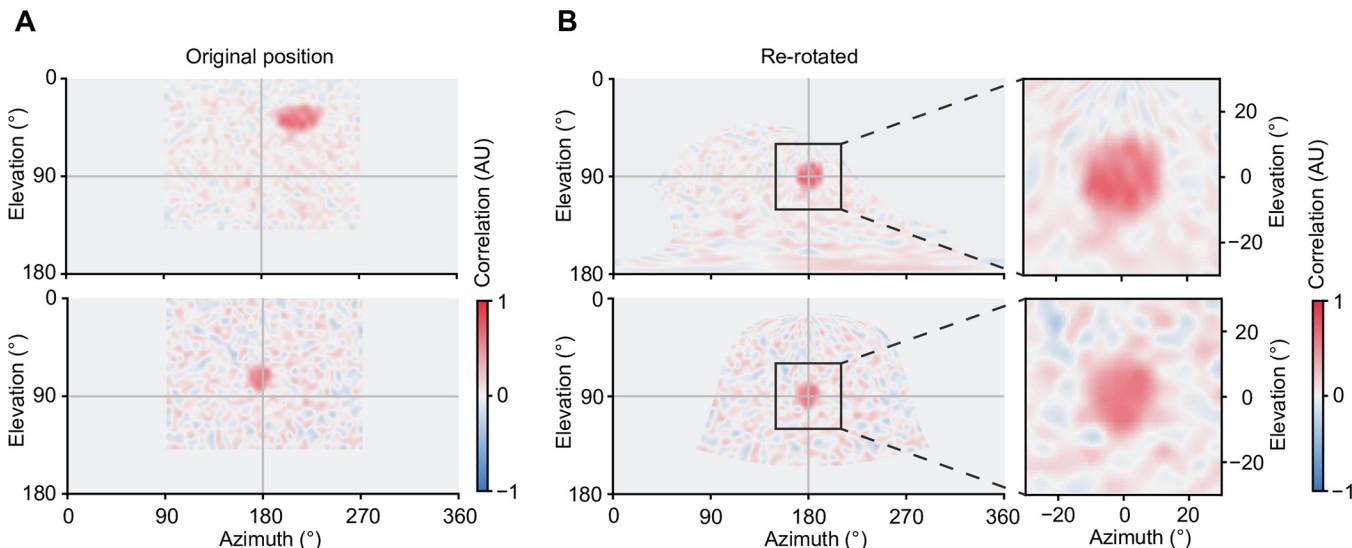

**Fig 4. Technical validation of spatiotemporal receptive field measurements. A** Equidistant cylindrical projection of the reverse correlation between input white noise and voltage signal of the photodiode. The "receptive fields" are displayed in their original positions. **B** Reverse correlation after rotation to the distortion-free center of the spherical projection map (left) and detailed views of the respective receptive fields after rotation for direct comparison (right).

## Electrophysiological validation

To validate the functionality of the Super Bowl in neurophysiological experiments, we performed *in vivo* whole-cell current clamp recordings of medulla intrinsic Mi9 neurons in tethered flies [17]. This experiment requires microscopic access to the brain, while retaining a wide field of view for the experimental animal. We recorded the membrane potentials of Mi9 neurons while presenting to the fly a binary white noise stimulus with a solid angle resolution of 2.8° over a period of 5 min. The spatial center of mass of each neuron's receptive field was determined and corrected by inverse rotation as in Fig 4 (Fig 5A). The largest deviations from the center were 38° in azimuth and 41° in elevation (Fig 5A). We estimated the time-averaged spatial receptive fields at two different time intervals (0 to 0.1 s and 0.2 to 0.6 s) and the temporal linear kernel at the center of the average spatial receptive field (Fig 5A). The full width at half maximum was ~10° (−3.5° to +6.5°) in azimuth and ~10° (−5.5° to +4.5°) in elevation, at both time intervals (Fig 5A I and 5A II), which is consistent with previous measurements [24]. The temporal linear kernel showed a negative correlation with a kernel density of approximately −7 at 0.05 s and a weak positive correlation a with kernel density of ~1 at 0.57 s (Fig 5A). This filter characteristic was absent in the temporal kernel of the photodiode (Fig 3F) speaking for distinct temporal filter properties of the medullary neuron. In addition, we tested a series of commonly used visual stimuli consisting of square-wave gratings and bright and dark edges moving at four different velocities. Unlike the photodiode signal (Fig 3E), the neurons' membrane potential responses to moving gratings were modulated in amplitude with the frequency of the stimulus (Fig 5B). The responses to moving bright and dark edges were aligned based on the location of the receptive fields of the respective cells (Fig 5C). They show step responses of opposite polarity to the stimulus contrast. In summary, the membrane potentials recorded from Mi9 neurons of flies stimulated using the Super Bowl screen were consistent with those from an established stimulation device [24, 25], with the difference that the data were acquired across a much wider field of view and with greater temporal precision.

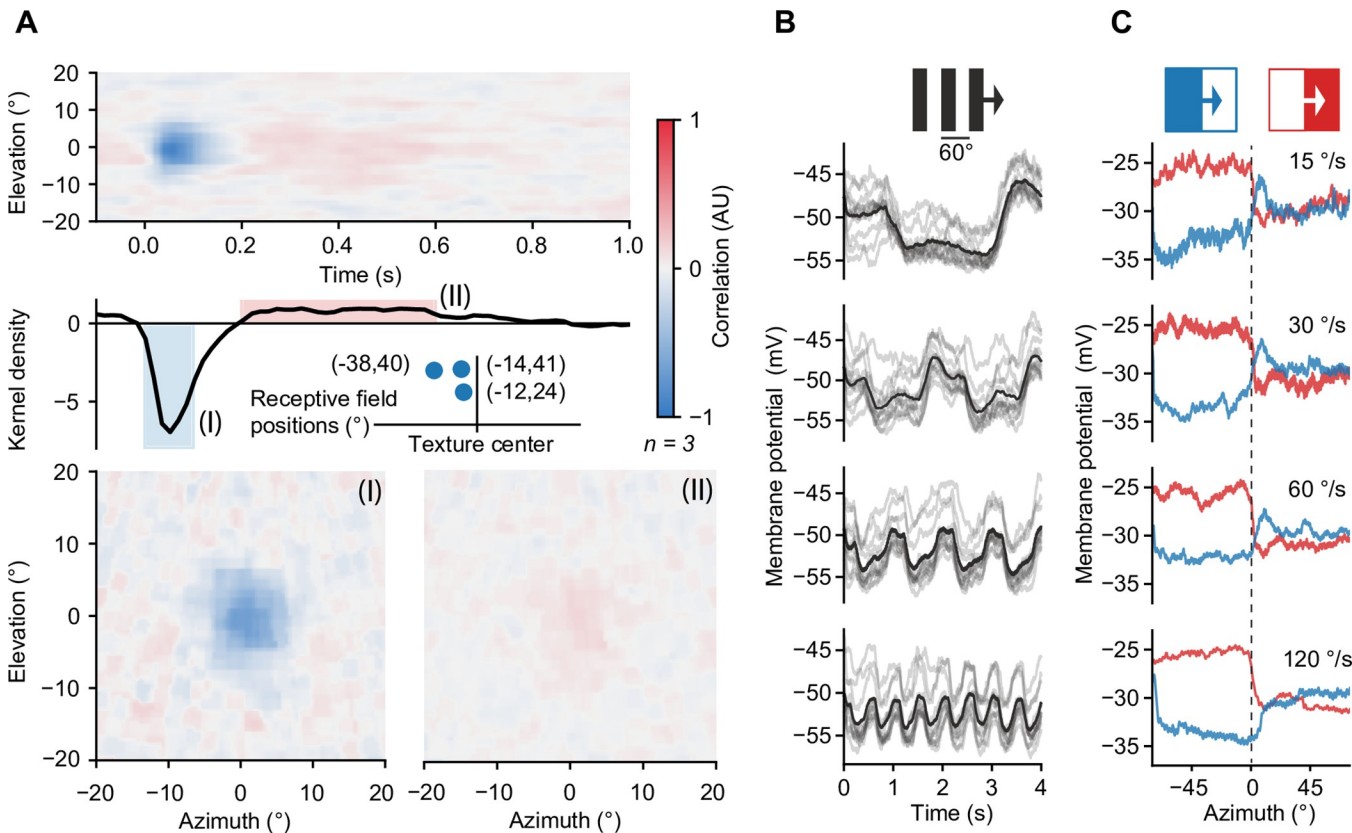

**Fig 5. Voltage responses of Mi9 neurons to visual stimulation of the bowl-shaped screen. A** Average spatiotemporal receptive field of three Mi9 neurons (top), temporal kernel density estimation at the center of the spatial receptive field (center), and relative location and coordinates of the centers of mass of the three receptive fields on the screen (center inset). (I) Time-averaged spatial receptive field in a time interval from 0 s to 0.1 s. (II) Time-averaged spatial receptive field in a time interval from 0.2 s to 0.6 s. **B** Average (black) and single-trail (gray) membrane voltage responses of one Mi9 neuron (n = 12 trails) to a 60˚-wide horizontal square-wave pattern, moving at four different velocities (15˚/s, 30˚/s, 60˚/s, 120˚/s). **C** Exemplary average membrane voltage responses of one Mi9 neuron (n = 3 trials) to bright and dark edges, moving at four different velocities (15˚/s, 30˚/s, 60˚/s, 120˚/s). Traces were aligned relative to the position of the receptive field.

## Behavioral validation

Besides the voltage responses of individual neurons, the behavior of the fly offers a second level of evidence for proper visual stimulation. The well-studied optomotor and fixation responses of flies [17, 26–28] provide clear behavioral readouts for validating the bowl-shaped screen. To observe these types of behavior, we added a tethered flight setup to the bowl-shaped screen (Fig 6A). To quantify the steering direction, a camera was placed above the fly to record the wingbeat envelope as a proxy of the wingbeat amplitude [29] (Fig 6A I). An additional camera, mounted on the side, was used to determine the horizon of the fly (Fig 6A II). The bowl-shaped surface of the screen made it possible to create virtual cylinders with different proportions surrounding the fly (Fig 6A). Like in the microscopic setting, the screen was positioned horizontally and aligned to the fly's horizon, so that the lower hemisphere of the fly's visual field could be stimulated.

To evoke an optomotor response, a virtual vertical cylinder, lined by a pattern of vertical stripes that extended from the horizon downward by 36˚, was rotated about the yaw axis of the fly head at a velocity of 60˚/s. Clockwise rotation gave rise to a positive deflection in the difference between left and right wingbeat amplitude, corresponding to a syn-directional steering command. Counterclockwise rotation of the virtual cylinder produced a negative deflection,

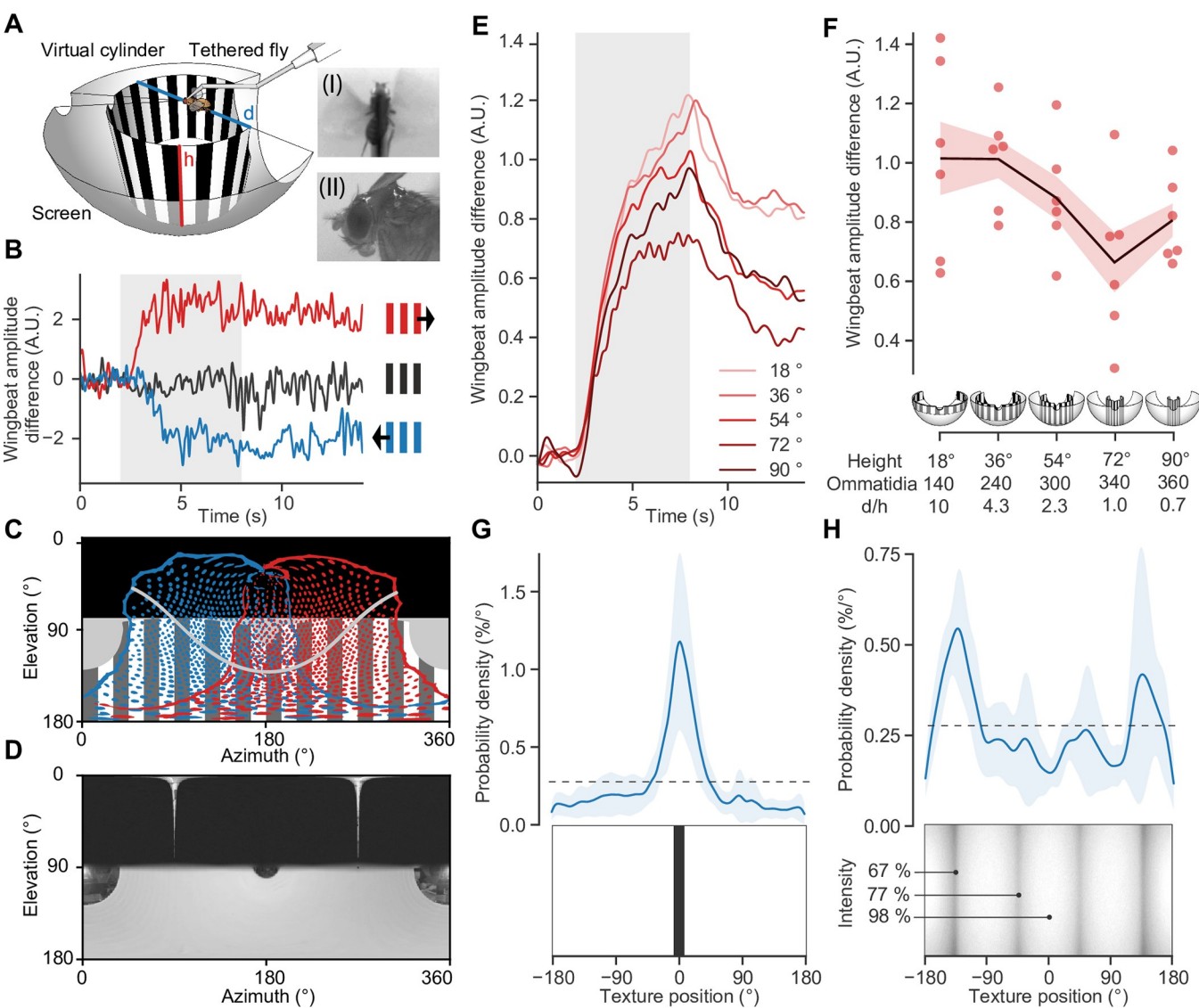

**Fig 6. Behavioural responses to visual stimulation of the bowl-shaped screen. A** View of the tethered flight setup, the bowl-shaped screen and a representation of a virtual cylinder with a height (h) of 54˚ and a diameter/height (d/h) ratio of 2.3. (I) View from above used to track the wing beat envelope. (II) View from the side used to determine the horizon. **B** Exemplary open-loop optomotor responses to a cylinder with a height of 36˚ and a spatial grating period of 30˚ rotating clockwise (red) or counterclockwise (blue) at a velocity of 60˚/s. The gray area indicates the time of stimulation. The black trace corresponds to a stationary grating. **C** Equirectangular projection of the stimulation field (grating) and visual field of the left (blue) and right (red) eye of the fly under microscopic constraints. The field of view was adapted from reference [20] and inclined by 45˚ to reflect the position of the fly head under the microscope. The black area is obscured by the fly holder. **D** Raw image of the screen brightness distribution measured with an 360˚ camera from the fly's point of view. The black area is covered by the flat fly holder. **E** Mean rectified wingbeat amplitude difference in response to rotating virtual cylinders of different heights (n = 6 flies, 50 trails). The gray area indicates stimulation duration, the different shades of red correspond to the respective cylinder heights. **F** Time-averaged optomotor responses from E at different cylinder heights (n = 6 flies, 50 trails). Red dots, individual flies; solid line, mean; red shading, standard error of the mean (SEM); d/h, diameter/height ratio of the virtual cylinder. The number of stimulated ommatidia was estimated based on reference [20]. **G** Closed-loop bar fixation experiment. Top: Probability of finding a 15˚-wide dark vertical bar at a certain position along the azimuth. Solid line, average probability density; shaded area, standard deviation (n = 6 flies, 54 trials); dashed line, uniform distribution. Bottom: Representative screen image with central bar. **H** Closed-loop fixation behavior in response to a texture with four dark edges. Top: Average fixation probability (top) relative to the texture (bottom). Solid line, average probability density; shaded area, standard deviation (n = 6 flies, 42 trials); dashed line, uniform distribution. The central luminance of two edges was 67%, that of the other two was 77%; the screen luminance in between edges was 98%.

while a stationary pattern led to no significant change in the wingbeat amplitude difference (Fig 6B). This result is comparable to previous measurements with other visual stimulation devices [17, 19, 29–35].

In contrast to other devices for visual stimulation under the microscope [4, 6, 10], the Super Bowl provides a larger field of view with a homogenous brightness distribution (Figs 2, 6C and 6D). To demonstrate the advantages of the new system, we designed two experiments that would otherwise not have been feasible. In the first experiment, we created a visual virtual environment where the fly is, again, at the center of a rotating hollow cylinder lined by a grating pattern. The wide field of view allowed us to simulate tall and thin as well as wide and flat cylinders without being restricted by the physical size of the screen. Virtual cylinders with diameter/height (d/h) ratios ranging from 0.7 to 10 were rotated clockwise and counterclockwise around the fly at a constant temporal frequency of 2 Hz while we measured the optomotor response of the fly. Trials with different cylinder dimensions and rotation directions were presented randomly and interspersed with stationary patterns. The response was strongest for flat and wide cylinders, which covered a rather small fraction of the field of view, and weakest for a cylinder of equal diameter and height (Fig 6E and 6F). The result of this experiment was unexpected, as the response strength was neither a linear function of the elevation angle covered by the cylinder, nor a monotonically increasing function of the number of stimulated ommatidia.

The second experiment concerns the influence of dark edges and brightness gradients—undesirable features of many stimulation devices, which our screen is virtually devoid of. Most visual stimulation devices suffer from some form of static brightness gradient. Brightness declines, for example, in the periphery of cylindrical projection screens, or at the gaps where multiple displays connect. If the visual contrast of the experimental stimulus approaches that of the static gradient, experimental artefacts are bound to confound the results. To expose the confounding influence of such subtle gradients, we resorted to the unmatched brightness uniformity and the wide field of view that the Super Bowl screen offers. We rendered an infinitely high cuboid that mimicked an arrangement of four inward-facing screens surrounding the fly. The inner surface of the virtual cuboid was rendered with non-perfect Lambertian properties, resembling real in-plane switching monitors, which produced realistic brightness gradients in the four edges. To test the flies' ability to perceive, and react to, subtle differences in luminance, two of the four edges were rendered marginally darker (67% vs. 77% luminance). In a closed-loop paradigm, the rotation of the cuboid was coupled to the left–right wingbeat amplitude difference of the fly, thus allowing the animal to rotate the virtual cuboid freely. Each fly was first tested with a single dark vertical bar. Consistent with previous results, flies steered toward the bar, keeping it in a narrow range in front of them for most of the time (Fig 6G) [5, 10, 36]. When facing the cuboid texture, the flies clearly preferred the darker edges (Fig 6H). The mean probability of a halt facing one of the two darker edges was twice as high as the mean probability of a halt facing one of the lighter edges. The implications of this experiment are twofold: First, it demonstrates that flies detect differences in luminance in the range of 10%, which are common when using conventional means of visual stimulation. Second, it demonstrates that these subtle luminance differences are sufficient to bias the animal's behavior.

## Discussion

The screen geometry of visual stimulation devices in neuroscience has important implications for the design of experiments and for the acquisition, the analysis, and the interpretation of functional data from the visual system. With a single observer at the center, a cylindrical screen is superior to a flat display, but observer-related perspective distortions remain. Super Bowl provides a virtual spherical texture, displayed on a bowl-shaped screen, free from projection-

related distortions, which are otherwise difficult to control. The bowl-shaped screen covers a much larger fraction of visible space with far fewer distortions than commonly used visual stimulation devices. Unlike systems with rear-projection screens [4, 6, 9, 13], it is not affected by light bleeding artefacts caused by subsurface reflections and scattering in the screen material. The peculiar geometry of the screen also facilitates image transformations, and bypasses the need for computationally expensive matrix calculations.

All crucial system components, including the screen, can be 3D printed. The overall cost of the system depends on the type of projector used, providing the user a maximum of cost control. The system is compact enough to allow for the recording of neural responses and animal behavior under a microscope (Figs 1E and 7A). The distance of the projector and the size of the screen can be adjusted to the needs of the experiment, for example, by incorporating a spherical treadmill to create an immersive virtual environment for walking flies (Fig 7). Alternatively, depending on the physical requirements, the screen can be trimmed and only a subset of it can be stimulated selectively (Fig 7B). The system was designed to be easy to manufacture and assemble. The provided software generates vector graphic files that are readable by a broad range of programs and can be used as a design basis for 3D printing and laser cutting. The processes of filling, painting and varnishing the screen are crucial to achieve Lambertian surface properties and an even brightness distribution, but they require neither special

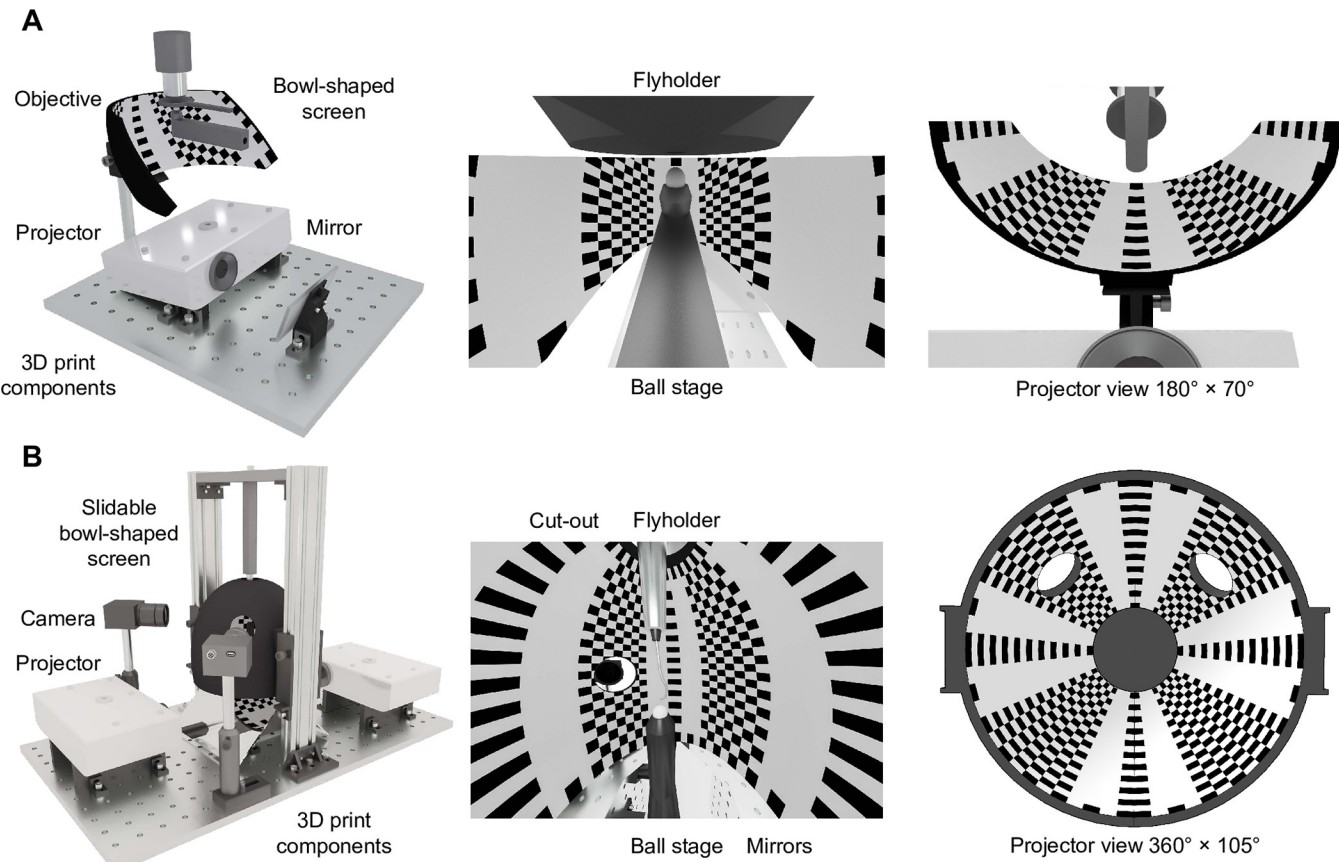

**Fig 7. Application concepts of the bowl-shaped screen. A** 3D rendering of a tethered walking behavior set-up, which also provides microscopic access to the brain. The field of view of the screen was designed in a way which enables closed loop bar fixation experiments, while recording neuronal activity. **B** 3D rendering of a tethered walking behavior setup, which creates an immersive virtual world around the fly. The system requires two projectors and allows tracking cameras to be installed at various locations using cut-outs.

equipment nor special practical skills. Projectors with identical throw ratios can be exchanged seamlessly, making it easy to incorporate the latest technological advances in spatial and temporal resolution.

The contrast achievable with this system depends on the type of projector used. Like in any projector-based system, black levels are inferior to those of light-emitting diode (LED) or organic light-emitting diode (OLED) screens. The maximum brightness, however, exceeds that of LED/OLED screens is in the range of (200–2500 cd/m2), together with measured contrast ratios of 1:50 to 1:300, this matches natural levels in cloudy daylight. As a consequence of the near-spherical geometry and the Lambertian surface properties of our system, the black level of a pixel—and hence the achievable contrast—is influenced by light reflecting off illuminated walls. This effect is akin to the way light falling into a room brightens both directly illuminated and shaded areas through room reflections. On white screens, this effect is more pronounced, but more predictable than reflection artefacts on glossy curved OLED screens. Linearity and gamma correction are widely adjustable. The colour depth exceeds that of LED-based systems [8] and is comparable to OLED and thin-film-transistor liquid crystal screens [5, 7].

The software for stimulus generation (https://github.com/borstlab/super_bowl_screen) enables researchers to create visual stimuli and entire visual environments. Accurately transformed projector output can be generated based on input from a game engine, a video, or a simple numeric array. The comparatively low computational demands of the software and fast online image transformations allow for the system to be used in closed-loop virtual reality settings that require fast response times. The performance can be increased if the calculation is carried out directly in a graphics library, like the Open Graphics Library (OpenGL).

Projector-based systems are compatible even with noise-sensitive electrophysiological experiments such as whole-cell patch clamp recordings (Fig 5). Local sources of electromagnetic noise, like the power supply unit or the projector itself, can be easily shielded. Some projectors can be operated in battery-powered mode, which increases the versatility even further.

Obtaining accurate measurements of the spatiotemporal receptive fields of neurons is challenging when using flat or cylindrical screens for at least three reasons: First, because receptive fields located at the periphery of the screen are subject to distortions, which lead to an overestimation of receptive field sizes (Fig 1A and 1B). Second, brightness and contrast tend to decline toward the periphery of a conventional screen. Together with perspective distortions, this reduces the correlation between luminance changes at each pixel and changes in neural activity, entailing longer acquisition times. Third, the surrounds of peripheral receptive fields are often obscured by the bezel. Therefore, it is common practice to include in the analysis primarily neurons with a receptive field at the center of the screen, which introduces a spatial picking bias and reduces the amount of analyzable data.

To investigate mechanisms of contrast normalization, especially the contribution of the visual surround on a neuron's receptive field [25, 37, 38], it is necessary to control luminance and contrast precisely across as much of the visual surround as possible. While the brightness of rear-projection screens [4, 6, 9, 13] and conventional LED displays [8] depends on the viewing angle and requires adjustment at the cost of dynamic range, the bowl-shaped screen features nearly uniform brightness (Fig 2) and only minimal perspective distortions (Fig 1A–1C). This allows for precise measurements of receptive fields of visual interneurons on a spherical map (Figs 4 and 5A), without the need for post-hoc corrections, as is the case when using cylindrical or flat screens.

The extensive field of view covered by the Super Bowl screen (Fig 6C) opens up new experimental possibilities, a glimpse of which we provided in a series of behavioral experiments (Fig 6). They revealed a counter-intuitive relationship between the extent of the visual stimulus and

the strength of the optomotor response (Fig 6E and 6F), which underscores the truism that the sum of all visual inputs rarely predicts the behavior of animals with a visual system as complex as that of the fly [39]. To those who seek to discover the neuronal mechanisms by which the fly estimates distance or height, an immersive virtual environment is essential. Uniform, well-controllable screen brightness, on the other hand, is a prerequisite for any study of perceptual decision-making; especially when operating close to psychophysical threshold, where subtle luminance differences start to matter. Using a closed-loop experiment, where the experimental animal was trapped inside a virtual cuboid, we showed that the fly perceived fine distinctions in brightness, comparable to those at the edges of conventional screens, and consistently chose to approach the darker edges (Fig 6H).

A visual system with an almost panoramic field of view, like that of *Drosophila*, should be stimulated with an immersive, distortion-free screen if we shall begin to understand its neuro-biology. Although spherical wide-field displays have been used in behavioral experiments [13], Super Bowl is the first system that can stimulate an extensive field of view while recording nervous activity under a microscope.

## Methods

Assembly instructions and a user manual for the Super Bowl screen are available at https://github.com/borstlab/super_bowl_screen and supporting videos are available at https://www.youtube.com/playlist?list=PLcB8ZWnb7EzSwOxb3ipZU5DP4hZYWpbc6.

### Analytical solution of bowl-shaped screen geometry

In two dimensions, the position of the fly is assumed to be at the center of the coordinate system, $Pos_{Fly} = \begin{bmatrix} 0 \\ 0 \end{bmatrix}$, and the position of the projector at a distance $d_{proj}$, i.e.

$$Pos_{Proj} = \begin{bmatrix} d_{proj} \\ 0 \end{bmatrix}.$$

The throw ratio $r_{throw}$ and the aspect ratio $r_{aspect}$ are constants and can be summarized to $r = r_{throw} \cdot r_{aspect}$. A projection offset of 100% (i.e.: upward projection) is required. The elevation angle $\alpha$ *with* $0 < \alpha < \pi$ in radians is used to parameterize the curvature of the bowl.

The viewing direction of each facet of the fly eye is defined as

$$\overrightarrow{\text{Fly}}(\alpha) = \begin{bmatrix} \cos(\alpha) \\ \sin(\alpha) \end{bmatrix}; \tag{1}$$

and the projecting direction of the projector is given as

$$\overrightarrow{\text{Proj}}(\alpha) = \begin{bmatrix} 1 \\ \alpha/\pi \ r \end{bmatrix} + \begin{bmatrix} d_{proj} \\ 0 \end{bmatrix}; \tag{2}$$

The two vectors intersect at point X, which has the coordinates:

$$\overrightarrow{\text{X}}(\alpha) = d_{proj} \frac{\alpha}{\alpha - \pi \ r \ tan(\alpha)} \begin{bmatrix} 1 \\ \tan(\alpha) \end{bmatrix}; \tag{3}$$

The vector $\overrightarrow{\text{X}}(\alpha)$ determines the curve of the bowl-shaped screen along the optical axis and one axis perpendicular to it. To parametrize the surface area in $\mathbb{R}^3$, a second variable $\beta = \beta \in \mathbb{R}$,

˚0<β<2π is needed for the azimuth. The surface S of the bowl-shaped screen is then given by the outer product:

$$\vec{S}(\alpha, \beta) = d_{proj} \frac{\alpha}{\alpha - \pi\, r\, tan(\alpha)} \begin{bmatrix} 1 \\ cos(\beta) \\ sin(\beta) \end{bmatrix} \cdot \begin{bmatrix} 1 \\ tan(\alpha) \\ tan(\alpha) \end{bmatrix}^{T}. \tag{4}$$

## Modeling and 3D-printing

The screen shape along the elevation axis was calculated using a custom-written script in Python v.3.7 (Python Software Foundation), using NumPy v.1.21.5, SymPy v.1.10.1, and Cairo v.1.16.0, based on the projector- and user-dependent input variables, such as distance, throw ratio, resolution, and field of view (https://github.com/borstlab/super_bowl_screen). Here, we specified a field of view ranging from 15˚ to 140˚ along the elevation axis. The automatically generated scalable vector graphics (SVG) file was imported into the computer-aided design program SketchUp (Trimble), where a 180˚ rotational extrusion along the optical axis of the path generated the final screen shape as a 3D surface mesh. After adding mounting connections to the drawing, a surface tessellation lattice (STL) file was exported for 3D printing. The screen was printed from black polylactide using fused filament fabrication on a Replicator 3D printer (UltiMaker). Comparable results were achieved using acrylnitril-butadien-styrol-copolymer on a Makerbot Method X (UltiMaker) or tough 2000 resin on a stereolithography-based Form 2 3D printer (Formlabs). Use of the latter significantly reduced the need for post-processing. Where a 3D printer is not available, online services such as Hubs (www.hubs.com), Materialise (www.materialise.com), Shapeways (www.shapeways.com), or Sculpteo (www.sculpteo.com) can be used to create the screen at competitive prices. Grooves in between filaments were filled using 2K General Purpose Body Filler (3M) to create a smooth texture. The surface was sanded using silicon carbide sandpaper with (grit 400) and painted with three layers of white paint (585036, Dupli Color). To reduce specular reflection and associated artifacts, a final layer of matte varnish (Matt varnish 850, Marabu) was applied which is a critical step to achieve near-Lambertian reflective properties. The surface properties of the coating were determined in a spectral range from 400 to 700 nm and may vary under illumination in the ultraviolet (UV) range. Irradiation with UV light (λ = 370 nm) did not give rise to any detectable fluorescence or pronounced changes in reflectance, suggesting that the coating could be used with UV projectors.

## Assembly

The dimensions, the distances and the optical light field of the projection geometry, and the design of the fly holders, were based on the vector graphic of the beam path (created in the previous step). In order to minimize the overall dimensions of the system, we installed two planar metal-coated mirrors (ME8S-G01–8", Thorlabs) perpendicular to each other and at a 45˚ angle relative to the projector (Fig 1E). Screen, projector, and mirrors were mounted on a frame made of matte black acrylic glass which was manufactured using a laser cutter (Speedy 360, Trotec). If a laser cutter is not available, the frame can also be 3D-printed, ordered from on-demand fabrication services (see above), or manufactured by conventional means in a workshop. To protect patch clamp electrodes from electromagnetic interference and to reduce the risk of electrostatic charging of the bowl-shaped screen, we mounted it on a grounded metal plate.

## Stimulus generation

Custom-written software in Python v.3.7 was used to generate visual stimuli, as well as to control their timing. For ease of use, stimuli were created using the NumPy v.1.21.5 package. Rotating textures were generated using just-in-time compilation using the JAX package (Google), to increase performance. The projection function was also based on the JAX package. Images were processed and presented using the OpenCV v.4.6.0 package and stimulus timing was implemented using the built-in time module of Python. After initialization, the software operated in a loop, which was run periodically with a stimulus-dependent rate of 80 to 120 frames per second. In each iteration, the input texture was recalculated (or rotated) based on the elapsed time. The resulting texture was then transformed, masked, and displayed by the projector. This procedure continued until the stop time point was reached. The entire procedure was processed in one thread; multithreading was not required, but would lead to additional gain in speed. The spatial resolution of the system is inevitably lower than the nominal projector resolution, as it depends also on the shape of the screen, the depth of field, and the projection distance. Given these constraints, and a projector resolution of $1280 \times 720$ pixels, our system had a measured spatial resolution of $0.5°$. To match the image quality, and to achieve a well-balanced compromise between computational efficiency and resolution, we used an input image of $720 \times 360$ pixels. This would suffice to stimulate with adequate resolution even insects with acute zones such as *Lucilia cuprina* or *Calliphora erythrocephala* whose interommatidial angles amount to $1.0°$ and $0.7°$, respectively [40].

The white noise stochastic stimulus used to determine the receptive fields, was updated online at a rate of 60 frames per second. The individual pixels corresponded to a visual solid angle on a virtual equirectangular cylindrical texture. For example, using a solid angle of $5°$, the field of view of the arena was divided into $36 \times 28$ pixels grid and the updating stimulus was displayed over 8 minutes. The intensities of the individual pixels were drawn randomly and were either 0% or 100% of luminance. Using a sliding reverse correlation, the impact of each stimulating pixel on the signal was reconstructed. Each reconstructed image thus corresponded to an equidistant cylindrical projection of the receptive field.

## Luminance measurements

To measure luminance profiles across the screen, we took two different approaches. First, we used a PM100D optical power meter (Thorlabs), whose sensor aperture was placed at the position of the observer. The angle of incidence was kept constant by rotating the sensor about its axis using a servo motor (RS-2, Modelcraft) and a gimbal tracking the position of a square spotlight along the azimuth of the screen. In addition, we placed a 360° camera (Fusion, GoPro) with a resolution of $3000 \times 3000$ pixels at the position of the fly and took a photo of the scene. To obtain temporal high-speed measurements, we used a 5-mm photodiode (L-7113GC, Kingbright) as a light sensor and acquired the voltage-signal generated by the photodiode using a 12-bit digital oscilloscope (WaveRunner HRO 66 Zi, LeCroy).

## Evaluation of screen geometries

To visualize the effect of screen geometry on the ratio of pixels per ommatidium, we generated computer-aided designs of a flat [5–7], a cylindrical [4, 8], and a bowl-shaped screen (this paper). In place of the insect, we added a Goldberg polyhedron model of the observer. The polyhedron contained 260 hexagonal faces of equal area (±1%) and, like any Goldberg polyhedron, 12 pentagonal faces, each of which with an area of approximately 66% of that of the largest hexagon. On average, one hexagonal face covered a solid angle of $14°$, equivalent to the angle covered by ~9 ommatidia of *Drosophila melanogaster*, a necessary approximation, given

that Goldberg polyhedron cannot assume any arbitrary number of faces. Twenty of the hexagonal faces were regular, the rest were irregular hexagons with aspect ratios ranging from 0.87 to 1.07.

The edges of the faces were projected radially onto the screen surfaces, using the open-source physically-based render engine LuxCoreRender (https://luxcorerender.org). The resulting renderings were vectorized and analyzed in Python v.3.7.13. Hexagonal projections were color-coded based on their screen surface area relative to that of a central, undistorted projection. To allow for a rigorous comparison between the different structures, a field of view covering 105˚ in azimuth and elevation was considered for all screen shapes (Fig 1A–1C). For visualization purposes, we used an orthographic view of the flat screen and a flattened orthographic projection of the cylindrical screen. Additional perspective distortions that would occur in the cylindrical setting under real projection conditions were neglected. The projection of the bowl-shaped screen was already a perspective projection, which did not require additional image distortion.

## Fly husbandry

*Drosophila melanogaster* were cultivated on a standard agar medium containing cornmeal, soy, molasses and yeast under a 12 h–12 h light–dark cycle at 25 ˚C and 60% humidity. All experiments were carried out on female flies bearing at least one wild-type allele of the *white* gene. Wild-type Canton-S flies were used for behavioral experiments and flies of the genotype P{R48A07-p65.AD}attP40, P{10XUAS-IVS-mCD8::GFP}su(Hw)attP5; P{VT046779-GAL4.DBD}attP2 were used to visualize Mi9 neurons during patch clamp experiments.

## Patch clamp recordings

Whole cell patch clamp recordings were performed *in vivo* as recently described [24]. Female flies were cold-anesthetized 2–24 hours after eclosion and fixed with soft thermoplastic wax on a custom-made polyoxymethylene mount. The dorsal part of the head was submerged in solution (pH 7.3) containing 5 mM TES, 103 mM NaCl, 3 mM KCl, 26 mM NaHCO$_3$, 1 mM NaH$_2$PO$_4$, 1.5 mM CaCl$_2$, 4 mM MgCl$_2$, 10 mM trehalose, 10 mM glucose and 7 mM sucrose (280 mOsM, equilibrated with 5% CO$_2$ and 95% O$_2$) and the left optic lobe was exposed by surgically removing cuticle, adipose tissue, and trachea using a stereomicroscope (S8 APO, Leica). Green-fluorescent somata were visually identified under an Axio Scope.A1 epifluorescence microscope (Zeiss). Patch pipettes (15–20 MΩ) were pulled from borosilicate glass capillaries using a PC-10 micropipette puller (Narishige) and targeted to the somatic cell membrane through a small incision in the perineural sheath. The pipette solution (pH 7.3) contained 10 mM HEPES, 140 mM potassium aspartate, 1 mM KCl, 4 mM MgATP, 0.5 mM Na$_3$GTP, 1 mM EGTA and 10 mM biocytin (265 mOsM). Membrane voltage signals were recorded at room temperature (21–23˚C) using a MultiClamp 700B amplifier (Molecular Devices), low-pass filtered, and sampled at 10 kHz. Voltage data were corrected for the liquid junction potential and analyzed using custom-written software in Python v.3.7. The resting potential was determined to be the most negative membrane potential recorded in the absence of any holding current. Cells with a resting membrane voltage more depolarized than –25 mV were excluded from further analysis.

## Tethered flight behavior

Female flies were cold-anesthetized two days after eclosion and attached to the tip of a needle at head and thorax using light-curing dental glue (3M Sinfony Opaque Dentin). The flight position was adjusted relative to the screen based on one camera on the side (Flir Chameleon3

CM3-U3-13S2C) and one camera above (Flir Flea3 FL3-U3-13S2M). The horizon of the fly eye was levelled before the start of each experiment; only flies with a well-aligned head (± 5˚) were included. Left and right wingbeat envelopes were recorded from above by acquiring images (656 × 524 pixels) of the tethered flying fly at 120 Hz and 8.3 ms exposure time. To minimize processing times and to enable behavioral experiments in closed loop, a digital wingbeat analyzer, analogous to a previous study [29], was implemented. Images were background subtracted and analyzed using two image masks covering the left and the right wingspan, respectively. The ratio of the mean pixel values of the two masks served as an indirect measure of the wingbeat amplitude and, thus, of the yaw torque [26, 29, 33]. Possible differences in illumination were compensated by weighting each mask individually. Wingbeat analysis (~8 ms) and image generation (~5 ms) were performed in sequence and completed within approximately 13 ms. The total closed-loop lag amounted to approximately 30 ms and was determined by the projector. The loop speed can be increased significantly by executing wingbeat tracking and image generation in two separate processes. This is particularly useful in combination with projectors that support higher frame rates.

## Optomotor responses

The virtual cylinders were lined by a vertical grating with a bar width of 15˚ and a height as specified (Fig 6E and 6F). Each stimulus phase was 14 seconds long, of which the first two were used as reference, followed by 6 seconds of stimulation and a 6-second-long post-stimulation period. Each trail consisted of three stimulus phases (clockwise, counterclockwise and static) in which the pattern was moved at 60˚/s in the respective direction or remained static. Cylinders of different diameter/height ratios and clockwise, counterclockwise and static phases were presented in random sequences. In open-loop experiments, the wingbeat amplitude signals were down sampled to 50 Hz, lowpass filtered with a symmetrical gaussian kernel ($\tau$ = 0.4 s) and the signals from the two sides were subtracted (left–right). The average wingbeat amplitude during the first two seconds, in the absence of visual motion, was taken as a baseline reference and subtracted from the signal. To obtain the total response strength of each fly, responses during clockwise and counterclockwise trials were rectified and averaged (Fig 6E). Time-averaged responses for each stimulus condition (Fig 6F) were calculated by averaging the wingbeat amplitude difference over a 6-s period starting two seconds after stimulus onset.

## Fixation behavior

At the beginning of each experiment, flies were allowed to accustom and become familiar with the closed loop system over a period of at least 5 minutes. During this phase, the weights of left and right wingbeat signals were determined and balanced, to prevent any unwanted rotation bias in slightly tilted flies. The feedback gain was set individually for each fly during this phase so that the fly was able to achieve rotation speeds of 0–140˚/s with its wingbeat amplitude. Once set, the gain was kept constant for all subsequent experimental conditions and trails. One trail of a fixation experiment consisted of initializing a texture at a random yaw position. Flies were then able to rotate the texture freely for 60 s during which the wingbeat amplitude difference was coupled to the yaw position of the texture. Each fly was subjected to both types of visual environment: a single vertical bar (Fig 6G) and the interior of an infinitely high cuboid (Fig 6H); trails were identically for both types of experiment. The position data of each trail was resampled to 50 Hz, and a probability density function was calculated for each trail independently. The individual trials of each fly were averaged to compute the fly's fixation response.

## Acknowledgments

We thank B. Zuidinga for help with fly husbandry and behavioral experiments and L. Fenk for discussions.

## Author Contributions

**Conceptualization:** Stefan Prech.

**Data curation:** Stefan Prech, Lukas N. Groschner.

**Formal analysis:** Stefan Prech, Lukas N. Groschner.

**Funding acquisition:** Lukas N. Groschner, Alexander Borst.

**Investigation:** Stefan Prech, Lukas N. Groschner.

**Methodology:** Stefan Prech.

**Project administration:** Lukas N. Groschner, Alexander Borst.

**Software:** Stefan Prech.

**Supervision:** Lukas N. Groschner, Alexander Borst.

**Validation:** Stefan Prech.

**Visualization:** Stefan Prech.

**Writing – original draft:** Stefan Prech.

**Writing – review & editing:** Stefan Prech, Lukas N. Groschner, Alexander Borst.

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
