## [Decision Letter · Decision Letter 0]

28 Feb 2024

PONE-D-24-00888Super Bowl, an open platform for visual stimulation of insectsPLOS ONE

Dear Dr. Prech,

Thank you for submitting your manuscript to PLOS ONE. I apologize for the delay - I am currently organizing the ICN2024 in Berlin, which is very labor intensive. After careful consideration, we feel that your manuscript has great merit and I am happy to report that only very minor changes were requested by the reviewers (see below). Therefore, we invite you to submit a revised version of the manuscript that addresses the points raised during the review process.

We look forward to receiving your revised manuscript.

Kind regards,

Mathias Wernet, PhD

Academic Editor

PLOS ONE

Journal Requirements:

Additional Editor Comments:

Dear Dr. Prech,

please excuse the delay in this matter - I am currently organizing the ICN2024 in Berlin, where I hope we will meet. I am happy to report that the reviews are very favorable and that only minor revisions and a few comments were requested.

Sincerely,

Mathias Wernet.

Reviewers' comments:

Reviewer's Responses to Questions

**Comments to the Author**

1. Is the manuscript technically sound, and do the data support the conclusions?

Reviewer #1: Yes

Reviewer #2: Yes

2. Has the statistical analysis been performed appropriately and rigorously? 

Reviewer #1: Yes

Reviewer #2: N/A

3. Have the authors made all data underlying the findings in their manuscript fully available?

Reviewer #1: Yes

Reviewer #2: Yes

4. Is the manuscript presented in an intelligible fashion and written in standard English?

Reviewer #1: Yes

Reviewer #2: Yes

5. Review Comments to the Author

Reviewer #1: The manuscript presented by Prech et al. describes a novel, versatile visual stimulation setup for insects and demonstrates its feasibility for studying response properties of visual neurons as well as visually guided behaviors. The authors provide a detailed description of the screen properties, highlighting advantages compared to other established visual stimulation methods and geometries in the field. The relatively small dimensions, large field of view and reduced computational cost due to the precalculated screen geometry make this setup a great option for many labs studying insect vision.

The manuscript is technically sound and written in an intelligible fashion. The data supports the authors conclusions with the statistical analysis being done in accordance with current scientific standards.

I only have a few minor comments:

Is the contrast the system can deliver comparable to other visual stimulation devices like LED matrices or liquid cristal devices?

Maybe the authors could add a few thoughts on this to the discussion.

While reading the manuscript it didn't become clear to me how well a fly is shielded from direct exposure to the projector beams. Eg would a fly placed in the fly holder in fig.1 be exposed to direct projector light from the back as well (creating large differences in the perceived intensity coming from the screen vs the projector)? Or would this effect be minimized by adding dark areas to the projection image according to a fly's position within the setup? I'd find it helpful if the authors could add a bit more clarification on this.

L16:...repeated presentations of the same stimulus tend to elicit different neural responses.

Please add reference

L452: dot is part of hyperlink and results in broken link.

Fig1a-c: what does the purple hexagon indicate? What do the arrows on the gray hexagon indicate?

Reviewer #2: I congratulate the authors on the thorough and meticulous work, clear presentation and an accessible manuscript that was a pleasure to read. The article will be an excellent guide. I have no major remarks and the manuscript can be accepted as is. However, to broaden the impact and guide the future builders, it may be useful to consider the following:

1. Is the intensity of projection system linear, easy to control, suitable for presenting arbitrary contrasts? What are the maximum and minimum contrast ratios? I know this largely depends on the projector used, but it is useful to remind people that it may be an issue.

2. Insect folks will be using UV capable projectors. In this case, the paint and varnish used here may not be appropriate. It would be good, but not strictly necessary, to know, if this particular combination is UV-reflective, UV-absorptive or perhaps even fluorescent.

3. A large piece of plastic such as the bowl in a dry lab may become charged with static electricity and even damage the ephys headstage. It could be wise to ground it just in case.

4. The angular resolution of Drosophila eyes is about isotropic across the visual field. Many other insects have regions named as acute zones with locally higher resolution. A projection system should match the highest angular resolution achieved in any compound eye. A reader might benefit from being aware of this.

Other comments:

L16 photoreceptors are very reliable and have extremely reproducible responses, the uncertainty probably accumulates downstream the visual pathway. [Please ignore this remark if you wish as the text flow is very nice here]

L43 I thought that the custom curved LED displays cover a substantial part of the visual field, IDK...

L257 change "velocity-depended" to "velocity-depending"

L441 add "to" in "comparable [to] those"

6. PLOS authors have the option to publish the peer review history of their article (what does this mean?). If published, this will include your full peer review and any attached files.

Reviewer #1: **Yes: **Thomas Frank Mathejczyk

Reviewer #2: No

---

## [Author Response · Author response to Decision Letter 0]

15 Mar 2024

We are grateful to our Editor and our Referees for the time and the effort they invested in the assessment of our work. The revised manuscript is improved considerably thanks to their advice. Please find a point-by-point response to the reviewers' comments in the attached file "Response to Reviewers".

---

## [Editor Report · Decision Letter 1]

27 Mar 2024

An open platform for visual stimulation of insects

PONE-D-24-00888R1

Dear Dr. Prech,

We’re pleased to inform you that your manuscript has been judged scientifically suitable for publication and will be formally accepted for publication once it meets all outstanding technical requirements.

Kind regards,

Mathias Wernet, PhD

Academic Editor

PLOS ONE
---

## [Editor Report · Acceptance letter]

8 Apr 2024

PONE-D-24-00888R1 

PLOS ONE

Dear Dr. Prech, 

I'm pleased to inform you that your manuscript has been deemed suitable for publication in PLOS ONE. Congratulations! Your manuscript is now being handed over to our production team.

Kind regards, 

on behalf of

Dr. Mathias Wernet 

Academic Editor

PLOS ONE